# DUBs Activating the Hedgehog Signaling Pathway: A Promising Therapeutic Target in Cancer

**DOI:** 10.3390/cancers12061518

**Published:** 2020-06-10

**Authors:** Francesca Bufalieri, Ludovica Lospinoso Severini, Miriam Caimano, Paola Infante, Lucia Di Marcotullio

**Affiliations:** 1Department of Molecular Medicine, Sapienza University, Viale Regina Elena 291, 00161 Rome, Italy; francesca.bufalieri@uniroma1.it (F.B.); ludovica.lospinososeverini@uniroma1.it (L.L.S.); miriam.caimano@uniroma1.it (M.C.); 2Center for Life Nano Science@Sapienza, Istituto Italiano di Tecnologia, Viale Regina Elena 291, 00161 Rome, Italy; 3Laboratory Affiliated to Istituto Pasteur Italia-Fondazione Cenci Bolognetti, Department of Molecular Medicine, Sapienza University, Viale Regina Elena 291, 00161 Rome, Italy

**Keywords:** ubiquitylation, DUBs, Hedgehog pathway, cancer, targeted therapy

## Abstract

The Hedgehog (HH) pathway governs cell proliferation and patterning during embryonic development and is involved in regeneration, homeostasis and stem cell maintenance in adult tissues. The activity of this signaling is finely modulated at multiple levels and its dysregulation contributes to the onset of several human cancers. Ubiquitylation is a coordinated post-translational modification that controls a wide range of cellular functions and signaling transduction pathways. It is mediated by a sequential enzymatic network, in which ubiquitin ligases (E3) and deubiquitylase (DUBs) proteins are the main actors. The dynamic balance of the activity of these enzymes dictates the abundance and the fate of cellular proteins, thus affecting both physiological and pathological processes. Several E3 ligases regulating the stability and activity of the key components of the HH pathway have been identified. Further, DUBs have emerged as novel players in HH signaling transduction, resulting as attractive and promising drug targets. Here, we review the HH-associated DUBs, discussing the consequences of deubiquitylation on the maintenance of the HH pathway activity and its implication in tumorigenesis. We also report the recent progress in the development of selective inhibitors for the DUBs here reviewed, with potential applications for the treatment of HH-related tumors.

## 1. The HH Signaling Pathway and Tumorigenesis: An Overview

The HH pathway is a mitogen and morphogen signaling, conserved from *Drosophila* to mammals. It plays a crucial role in organogenesis and central nervous system (CNS) development [1,2]. In post-embryonic stages, HH signaling regulates tissue homeostasis and repair, modulating the specification of the adult stem cells [3,4]. Several studies have highlighted similarities and divergences between *Drosophila* and mammals HH signal transduction (Figure 1A,B). Both in flies and in vertebrates the HH pathway activation is finely orchestrated by two membrane receptors: the multi-pass transmembrane protein Patched (Ptc/PTCH) and the heptahelical transmembrane co-receptor Smoothened (Smo/SMO). In *Drosophila*, in absence of the Hh ligand, Ptc keeps off the signaling by directly affecting Smo activity and preventing its accumulation on the plasma membrane. In this state, Costal-2 (Cos2; Costa-FlyBase), a kinesin family protein, Fused (Fu), a serine-threonine kinase and the Suppressor of fused [Su(fu)] inhibit the bifunctional transcription factor *Cubitus interrupts* (Ci), endowed of both repressor and activator domains. The full-length Ci protein is proteolytically processed by the Skp1-Cullin1-Slimb (SCF^Slimb^) ubiquitin ligase complex, in a truncated form (Ci^R^) that acts as transcriptional repressor of Hh target genes when translocated into the nucleus (Figure 1A) [5,6].

In mammals, three ligands belonging to the HH family are secreted: Desert hedgehog (DHH), Indian hedgehog (IHH) and Sonic hedgehog (SHH). The proteins, encoded by three paralogous mammalian genes, share high similarity in the affinity with HH-binding proteins. SHH is mostly expressed in brain cells and implicated in central nervous system (CNS) development, while IHH modulates chondrogenesis, and DHH regulates spermatogenesis and nerve-Schwann cell interactions [7].

A peculiar characteristic of HH signal transduction is the role of the primary *cilium*. This organelle is a microtubule-based protrusion of the cell membrane that coordinates protein trafficking events, recruits and stabilizes a regulative dynamic network among the core of HH components [8]. 

The complexity of HH signaling in vertebrates is also provided by the GLI zinc-finger transcription factors, the final effectors of the pathway (Figure 1B). Three GLI members have been identified, GLI1, GLI2 and GLI3: GLI1 acts exclusively as activator, instead GLI2 and GLI3, which have an N-terminal repressor domain, can work either as repressors or activators [9]. The balance between activator and repressor forms is widely ruled by SUFU, an essential negative regulator that controls HH signaling through its direct interaction with GLI factors [9,10].

When the HH pathway is off, phosphorylated GLI1 is recognized by the Skp1-Cullin1-βTrCP (SCF^βTrCP^) ubiquitin ligase complex and degraded by proteasome system, whereas GLI2 and GLI3 undergo a proteolytic process that converts them into cleaved transcriptional repressor forms. Otherwise, the binding of mature HH ligand to PTCH receptor releases the inhibition exerted on SMO, resulting in its activation and translocation into the *cilium*. These events lead to the nuclear localization of GLI activator forms where they induce the expression of HH-target genes, which include *GLI1* itself, thus triggering a positive feedback loop that amplifies the signal [11,12]. 

The HH pathway output is tightly regulated at multiple levels by different post-translational modifications, such as phosphorylation and ubiquitylation [13,14,15]. The pattern of GLI phosphorylation triggered by the protein kinase A (PKA), the casein kinase 1 (CK1α) and the glycogen synthase kinase 3 (GSK3β) establishes multiple states of GLI activity and ultimately influences the HH transcriptional program [16]. The sequential phosphorylation of GLI proteins leads to the recruitment of the SCF^βTrCP^, thus promoting GLI ubiquitylation and proteasome-mediated processing, as also described for its homolog Ci in *Drosophila* [17]. 

The ubiquitin-mediated processes of GLI factors are also triggered by other E3 ligases, such as the RING Cullin3-HIB/Roadkill/SPOP complex, the acetyltransferase/E3 ligase PCAF (P300/CBP-associated factor), and the HECT E3 ligase Itch. Importantly, Itch controls HH signaling by distinct routes: it mediates regulatory events on SUFU and proteasome degradation of GLI1 and PTCH1 by the interaction with the adaptor proteins β-arrestin2 and Numb, respectively [18,19,20,21,22,23,24,25].

In the last years, post-translational modifications have also been described to control SMO activity. As GLIs, SMO is regulated, in response to HH stimuli, by PKA/CK1-mediated phosphorylation in *Drosophila* and GRK2/CK1α in mammals, and downregulated by ubiquitin-mediated endocytosis and ubiquitin-dependent lysosome or proteasome degradation [26]. In *Drosophila*, Smo ubiquitylation and trafficking on cell surface is regulated by the HECT E3 ligases Smurf and Herc4, and the E3 ligase complex formed by Cullin4 and DNA-damage-binding protein 1 (DDB1), recruited by Smo through the β subunit of trimeric G protein (Gβ) [27,28]. Moreover, in mammals HERC4 has been described as tumor suppressor in non-small cell lung cancer (NSCLC) able to control SMO protein stability [29]. 

Given the essential role of HH signaling for a proper development, mutations in its key players cause congenital malformations [30]. An uncontrolled and permanent activity of the HH pathway is also associated to various human cancers such as basal cell carcinoma (BCC), medulloblastoma (MB), gliomas, pancreatic, colorectal, prostate, lung, and breast cancers (Figure 2) [31,32,33]. Indeed, aberrant HH activation involves and triggers pro-tumorigenic events, such as proliferation, survival, angiogenesis, migration and epithelial-mesenchymal transition (EMT) [34], thus affecting every step of carcinogenesis, from early development to metastatic progression [31,32]. 

Hyperactivation of HH signaling can occurs through either ligand-independent or ligand-dependent mechanisms. Tumorigenesis is ligand-independent when the pathway is constitutively activated in the absence of ligand via mutations in HH signaling components. Loss-of-function mutations in *PTCH* or *SUFU* or gain-of-function mutations in *SMO*, as well as *GLI1* overexpression or *GLI2* amplification have been identified in BCC, a common human skin cancer, and in MB, a highly malignant pediatric brain tumor [35,36,37,38,39]. Depending on the type of HH ligand release, two mechanisms of ligand-dependent pathway hyperactivation have been described in cancers, generating a tumor-stromal crosstalk [40]. Ligand-dependent autocrine/juxtacrine secretion occurs when the HH ligand is profusely released and caught by the same tumor cells, thus activating the pathway. Tumors that arise from this condition may display HH ligand overexpression or high levels of PTCH1 and GLI1 [41,42,43]. Alternatively, a paracrine secretion of HH ligand by tumor cells can induce the activation of the HH pathway in stromal cells of tumor microenvironment. As consequence, the stroma secretes paracrine growth signals to induce tumor growth [44]. For instance, in prostate cancer specimens, the expression of HH was detected in the tumor epithelium, while GLI1 expression was found in the tumor stroma cells, suggesting their paracrine crosstalk [45]. Moreover, this mechanism of HH signaling activation can work in a reverse paracrine manner in which cancer cells take the HH ligand released by stromal cells. For example, HH ligand released by bone marrow, nodal and splenic stroma can activate the HH pathway and maintain the survival of B and plasma cells in hematological malignancies [46]. Interestingly, HH-producing microenvironment is required for GLI activation in gliomas [47].

Of note, HH signaling also regulates the expression of the stemness genes *Nanog* and *Oct4*, thus participating in the formation or maintenance of cancer stem cells (CSCs) responsible of tumor initiation, relapse and drug resistance [48,49,50]. For all these reasons, the HH pathway is emerged as an attractive druggable target for anti-cancer therapy. A various number of SMO antagonists, able to block the pathway at upstream level, have been identified and patented. Some of them, vismodegib and sonidegib, and recently glasdegib, have been approved by the Food and Drug Administration (FDA) for the treatment of BCC and Acute Myeloid Leukemia (AML), respectively [34]. Many others, such as GANT61 and GlaB, have been designed targeting GLI1, the downstream effector of HH signaling, and have shown efficacy in preclinical study [34,51,52]. The major issue in employment of HH-inhibitors is the recurrence of drug-resistance mutations or alternative mechanisms of activation. Consequently, multi-target therapy is emerging as a promising strategy for the treatment of HH-dependent cancers. The best approach envisioned so far is the development of further inhibitors, or the identification of additional regulators of the HH pathway that could be targeted in tumorigenesis.

## 2. Ubiquitylation Process

Ubiquitylation dictates the fate and function of most cellular proteins increasing the complexity of the proteome. This modification is a dynamic and tightly regulated post-translational event with many distinct outcomes affecting protein stability, localization, interactions, and activity.

Ubiquitin (Ub) is a small globular protein consisting of 76 amino acids encoded in mammals by four different genes (*UBB*, *UBC*, *RPS27*, and *UBA52*) that ensure high cellular Ub levels [53]. Ubiquitylation is a multi-step process orchestrated by an enzymatic cascade that relies on Ub and three different enzymes: Ub-activating (E1), Ub-conjugating (E2), and Ub-ligating (E3) [54]. During the catalytic reactions, Ub is activated in an ATP-dependent way by an E1 enzyme, subsequently transferred to the active cysteine (Cys) residue of an E2 enzyme via a trans-(thio) esterification reaction, and finally attached with an isopeptide bond to a substrate by an E3 enzyme (Figure 3A). In humans, two E1s, around 30 E2s and over 600 E3s have been identified [55,56]. The latter are the major determinants and provide specificity for substrate recognition. Based on their functional domains and on the mechanism of catalysis, E3s are divided into three main families: the Really Interesting New Gene (RING), the Homologous to the E6-associated protein Carboxyl-Terminus (HECT) types, and RING-between-RING (RBR), which can be considered a RING-HECT hybrid [57,58]. Each class of E3 ligases can create Ub linkages of different length and architecture. The transfer of the Ub moiety to substrate occurs through the formation of the covalent bond between α-carboxyl group of the terminal glycine (Gly) residue of Ub and, commonly, ε-amino group of an internal lysine (Lys) residue of the substrate. Of note, for a subset of substrates the attachment of Ub may interest their N-terminal residue, a process known as N-terminal ubiquitylation [59], or serine and threonine residues, further expanding the complexity and the biological relevance of this process. In this regard, Ub modifications of a target protein occur in various forms: attachment of a single Ub moiety on a single substrate residue (monoubiquitylation), a single Ub on multiple residues (multi-ubiquitylation), or additional Ub molecules to initial Ub yielding an ubiquitin chain (poly-ubiquitylation). Typically, mono- and multiubiquitylation regulate endocytosis, signal transduction, DNA repair, and often result in changes in the cellular localization and protein activity [60,61,62]. By contrast, polyubiquitylation is the most abundant modification that controls protein homeostasis. Indeed, the polyubiquitylated target substrates are recognized by the 26S proteasome, a multiprotein complex, that degrades the proteins into small peptides and releases the Ub for cyclic utilization [63]. Besides regulating protein degradation, polyubiquitylation brings different functional consequences depending on Ub chain linkage-type [64]. Ub has seven Lys residues (K6, K11, K27, K29, K33, K48, and K63) that may serve as polyubiquitylation points. Depending upon the Lys used, length of the chains and linkage type, distinctive forms of Ub chains may be achieved to drive the fate of target proteins [65]. Lys48-linkage targets protein for proteasome-dependent degradation, whereas Lys63-linkage is associated to regulative processes, including trafficking, protein localization, protein-protein interaction; the biological significance of other Ub modifications is still largely unclear [66]. Further complexity is provided by Ub modifications (i.e., phosphorylation, acetylation, sumoylation) and by the linkage of Ub to other Ub-like proteins (i.e., NEDD8, SUMO), creating a multitude of distinct signals. The combination of all these parameters has been referred as the “Ub code” [65]. The Ub code governs the fate of the targeted substrates by modulating their interactions with many other proteins that incorporate Ub-binding domains and determine their accessibility to deubiquitylating enzymes (DUBs), a family of protease conserved from yeast to humans [67].

## 3. Deubiquitylating Enzymes: Functions and Classification

Like other important post-translational modifications, ubiquitylation is a dynamic and reversible process counteracted by DUBs activity [65]. DUBs are proteases that hydrolyze isopeptide or peptide bond removing Ub conjugates from substrates and disassembling anchored Ub chains (Figure 3B) [65,68]. DUBs may remove Ub moieties from the distal end or through the cleavage within chains in two distinct ways: i) via direct interaction with specific substrates; ii) through selective recognition for particular Ub chain architecture. Both chain length and linkage type may drive the choice of the target proteins. Importantly, linkage selectivity may occur within the catalytic domain or through the cooperation with Ub-binding domains within DUBs or their interaction partners [68].

Given their crucial role in opposing E3 ligases function, DUBs control protein homeostasis and activities, and are implicated in the regulation of various physiological and pathological processes, such as development, metabolism, immune response and tumorigenesis.

Currently, 99 cellular DUBs have been identified and are classified into six main families depending on distinct catalytic domains: the largest group ubiquitin-specific proteases (USPs), ubiquitin C-terminal hydrolases (UCHs), ovarian tumor proteases (OTUs), JAD/PAD/MPN-domain containing metalloenzymes (JAMMs), Machado-Joseph disease domain proteases (MJDs or Josephins) and motif interacting with Ub-containing novel DUB family (MINDYs) [69,70]. Unlike of the JAMM family, classified as a zinc-dependent metalloproteinase, the other DUBs classes are cysteine proteases. Available data indicate that each family may display linkage or substrate preferences. For instance, OTU family exhibits linkage type specificity, whereas USP group members show differences in catalytic rate constants [68,71,72]. Studies aimed at defining the abundance of individual DUBs suggest that those with constitutive functions show high copy number, while DUBs with peculiar roles are the rarer forms [70]. Different approaches used to determine the intracellular localization of the DUBs allowed highlighting that subsets of these proteases show particular association with subcellular compartments. Although many DUBs are nuclear, several USP members localize to defined structure including plasma membrane, microtubules, endosome, and endoplasmic reticulum (ER) [73].

To date, a growing body of evidence indicated that DUBs can act as oncogenes or tumor suppressors emerging as a promising class of therapeutic targets. For these reasons, many efforts are devoted to the development of highly selective DUBs inhibitors for anti-cancer therapies.

## 4. Oncogenic DUBs Involved in the Regulation of the HH Pathway

### 4.1. DUBs Acting on SMO

SMO is the main upstream signal transducer of the HH pathway in both insects and vertebrates. SMO is classified as an atypical G protein-coupled receptor (GPCR), since it possesses stereotypical GPCR functional domains: seven transmembrane domains (TM), an intracellular C-terminal tail, an amino-terminal cysteine rich domain (CRD), three extracellular and three intracellular loops (ECL and ICL) [74,75].

The molecular mechanisms that induce SMO activity in response to the activation of the HH pathway represent a crucial question in the understanding of HH signal transduction. In *Drosophila*, activated Smo accumulates in the plasma membrane [76,77], while in vertebrates it translocates into the primary *cilium*, a small protruding organelle in which all the key components of HH signaling are enriched [78,79]. Post-translational modifications regulate Smo activity. At present, the positive role of phosphorylation on Smo subcellular trafficking and activation is well established: in *Drosophila* protein kinase A (PKA) and casein kinase 1 (CK1)-mediated phosphorylation promotes Smo cell surface localization [80,81,82,83], whereas in vertebrates GRK2 and CK1α-dependent phosphorylation of SMO C-tail has been found to be pivotal for its ciliary accumulation [83]. In the last years, the role of ubiquitylation as negative modulator of Smo, due to the involvement in its endocytosis, trafficking and degradation has increasingly emerged [26,84]. 

#### 4.1.1. USP8

Ubiquitin-specific protease 8 (USP8) is a multi-domain deubiquitylating enzyme with pleiotropic functions. Besides its canonical role in protein trafficking and receptor tyrosine kinase degradation, USP8 controls other biological processes, such as endosomal sorting, mitochondrial quality control, ciliogenesis and apoptosis [85]. Indeed USP8 was found to deubiquitylate the E3-ubiquitin ligase Parkin, involved in autophagy of dysfunctional mitochondria, the HIF1α protein, important for endosome trafficking-mediated ciliogenesis, and c-FLIP a master anti-apoptotic player [85]. Recently, the involvement of USP8 in the regulation of Hh signaling, through the stabilization of Smo, has been described.

Two independent studies have demonstrated that the absence of Hh ligand induces both the poly- and monoubiquitylation of Smo, leading to its endocytosis and degradation both by the lysosome- and proteasome-mediated pathway, in order to keep Hh signaling off [26,84]. Conversely, upon ligand stimulation, Smo is deubiquitylated and hence accumulated on the cell surface, where it becomes activated [84]. By using an in vivo RNAi screen that targeted *Drosophila* DUBs, Xia and colleagues identified USP8 as a deubiquitylase that prevents Smo ubiquitylation and is required for Hh-induced cell surface accumulation of Smo, thus increasing Hh signaling activity [84]. 

Similar results have been obtained in NIH3T3 cells, suggesting a conserved mechanism that controls SMO in mammals. Moreover, the authors sustain a link between phosphorylation and ubiquitylation in controlling Smo activity. Indeed, Hh promotes the interaction of USP8 with Smo aa625–753, the residues phosphorylated by PKA and CK1, showing that phosphorylation of Smo induces the formation of Smo/USP8 complex and amplifies Hh stimulation (Figure 4A,B). Parallelly, the sumoylation of Smo at K851 induced by Hh, recruits USP8 to inhibit Smo ubiquitylation and degradation, leading to its cell surface trafficking and amplifying the Hh pathway activity, both in *Drosophila* and mammals [86]. These data stand USP8 as a positive regulator in the HH pathway, able to prevent SMO localization to early endosomes, promoting its stability [84].

#### 4.1.2. UCHL5/UCH37

A similar role to USP8 has been described by Zhou et al. for the deubiquitylase UCHL5 able to increase the protein stability and the cell membrane accumulation of Smo [87]. UCHL5 (also known as UCH37 in mammals) is a deubiquitylase involved in the regulation of several substrates (i.e., type I TGF-β receptor, E2 promoter binding factor 1) [88,89] and is formed by an N-terminal UCH and a C-terminal extension domains (Figure 4B) [90]. In *Drosophila*, the UCH region of UCHL5 binds Smo C-tail [87]. Through its C-terminal fragment, UCHL5 recruits Rpn3, a proteasome subunit that increases UCHL5 deubiquitylating activity and forms a trimetric complex with Smo, thus reducing its ubiquitylation. Moreover, UCHL5 inhibits the interaction of Smo with the hepatocyte growth factor-regulated tyrosine kinase substrate (Hrs), known to promote Smo ubiquitylation [91]. Interestingly, ubiquitylation assays performed in knockdown conditions of *UCHL5* and *USP8* demonstrated that this two DUBs cooperate to deubiquitylate and stabilize Smo [87]. The activation of the Hh pathway does not affect the expression levels of UCHL5, but increases the affinity between UCHL5 and Smo, stabilizing the receptor with its consequent localization at the cell membrane [87]. Importantly, this mechanism is conserved in mammals through its homolog UCH37 [87]. Many evidence show that UCH37 is upregulated in a wide spectrum of tumors, suggesting its potential oncogenic role in tumorigenesis [92,93,94].

Although the negative role of Smo ubiquitylation in the control of Hh activity is well established, only recently the E3 ligases involved in this process have been identified in *Drosophila*, and include Uba1, Cul4-DDB1, Smurf, and Herc4 (Figure 4A) [26,27,28,84,95,96]. In particular, recent findings displayed that the HECT E3 ligase Herc4 binds Smo and mediates its mono- and polyubiquitylation at multiple Lys residues, thus promoting its lysosome and proteasome degradation. The interaction between Smo and Herc4 is inhibited by Hh that prevents Herc4-mediated Smo ubiquitylation in a manner independent of PKA-primed phosphorylation [95]. Importantly, Herc4 interacts with USP8 and UCHL5 and their overexpression almost abolishes Herc4-mediated Smo ubiquitylation, by blocking the association between Herc4 and Smo [95]. In mammals, HERC4 binds SMO and induces its degradation. In human NSCLC, *HERC4* knockdown activates HH signaling and promotes NSCLC cell proliferation thus standing as a tumor suppressor [29]. 

Multiple E3 ligases and DUBs are involved in the fine regulation of SMO stability and trafficking, and the perturbation of their function could alter the HH pathway activity. In particular, given the positive role of DUBs in controlling HH signaling, they emerged as a potential drug target for HH-related tumors. 

### 4.2. DUBs Acting on GLI Factors

GLI zinc finger transcription factors are the main effectors of HH signaling. Both SMO-dependent and independent HH pathway activation culminate with the nuclear translocation of GLIs, promoting the expression of HH target genes. GLIs function is widely ruled by post-translational modifications. In particular, GLI ubiquitylation is orchestrated by several E3 ligases belonging both to the RING (such as SCF^βTrCP^ and Cullin3-HIB/Roadkill/SPOP [17,97,98]) and the HECT (Itch) families [24,99], and the non-canonical E3 ligase PCAF [25,100]. This modification leads to proteolytic cleavage of GLI2 and GLI3 factors [97,98] or massive degradation especially for GLI1 protein [17,23,24].

#### 4.2.1. USP7

Ubiquitin-specific protease 7 (USP7, also called Herpes virus-associated protease, HAUSP) is the first identified deubiquitylase isolated as a partner of the herpesvirus protein [101]. USP7 is a cysteine peptidase primarily located in the nucleus where it controls the stability of multiple proteins involved in the Zhou and colleagues described USP7 as positive modulator of HH signaling in flies and vertebrates. Indeed, Usp7 in *Drosophila* and its homolog HAUSP in mammals antagonize multiple E3 regulation of DNA damage response, transcription, epigenetic control of gene expression, immune response, and viral infection. Indeed, among the many substrates of USP7 are included the tumor suppressor proteins p53 and PTEN, the oncoproteins C-Myc and N-Myc, the transcription factors Foxp3 and FOXO family members, the DNA methyltransferase 1 (DNMT1), the checkpoint kinase 1 (CHK1) and viral proteins, such as EBNA1 and ICP0 [102]. 

In mouse *Usp7* knockout is lethal [103,104], while in human its mutations and deletions have been recently identified in children suffering from neurodevelopmental disorders [105]. ligases function to maintain the HH pathway activity [106]. In particular, upon Hh treatment Usp7 interacts with Ci through multiple P/AxxS motifs and increases its protein stability [106]. Usp7 localizes in both cytoplasm and nucleus and counteracts respectively SCF^Slimb^ and Hib-Cul3-mediated Ci degradation (Figure 5A) [106]. 

Similarly, USP7 binds all GLI factors in mammals, and these interactions are favored by HH and hindered by SUFU [106]. USP7 stands as positive regulator of HH signaling that stabilizes GLIs protein levels by antagonizing either the Itch-dependent degradation of GLI1 [24,99], and the SPOP/CUL3-dependent degradation of GLI2/GLI3 (Figure 5A) [107,108]. 

*Usp7* knockout in mouse cause embryonic lethality at embryonic days (E) 6.5–7.5 [103], while in human USP7 shows an oncogenic role in neoplastic diseases such as NSCLC, human prostate and liver cancers [109,110]. Zhan and collaborators also investigated the effects of USP7 modulation on human MB, the most common pediatric tumor of the cerebellum [111]. About 30% of all MBs arises from HH signaling aberrant activation (HH-MBs) [112]. *USP7* depletion inhibits the proliferation rate, the migration capability and the invasiveness of human HH-MB Daoy cells due to the decrease of GLIs protein levels and of HH target genes transcription [113]. The treatment of Daoy cells with the USP7 inhibitors, P5091 and P22077, blocks their proliferation and metastasis [113] standing USP7 as potential druggable target in SHH-MBs.

#### 4.2.2. USP48

Ubiquitin-specific protease 48 (USP48) contains an ubiquitin C-terminal hydrolase (UCH) domain, required for its catalytic activity, and an ubiquitin-specific proteases (DUSP) domain mostly involved in protein-protein interaction (Figure 5B) [114]. Several substrates of USP48 have been recently identified, such as the tumor necrosis factor receptor-associated factor 2 (TRAF2) related to JNK pathway, the histone H2A and RelA, a member of the avian reticuloendotheliosis/NF-κB transcription factors family [115,116,117]. Moreover USP48 is a novel binding partner of Mdm2, promoting its stability with a deubiquitinase activity-independent mechanism [118]. USP48 is expressed in almost all human tissues [119] and is upregulated in malignant melanoma [120]. Zhou and co-authors recently highlighted the USP48 involvement in HH signaling regulation and its role as promoter of glioblastoma cell proliferation and tumorigenesis [121]. USP48 and GLI1 co-localize in the nucleus, interacting through the N-terminal sequence of GLI1 and the C-terminal DUSP domain of USP48 [121]. This interaction protects GLI1 from proteasome-dependent degradation thus increasing its protein stability (Figure 5A). The specific function of USP48 on GLI1 promotes the proliferation and the colony formation of glioma cells in vitro. Moreover, its depletion abrogates the tumor formation and extends the survival rate of orthotopic glioblastoma mouse models in vivo [121]. Zhou and colleagues sustained a positive feedback loop by which HH signaling activates USP48 through the binding of GLI1 to *Usp48* promoter. Of note, USP48 and GLI1 expression levels directly correlate in human glioblastoma specimens, and they are linked to tumor malignancy grade. This evidence underlies the relevance of USP48-GLI1 regulatory axis for glioma cell proliferation and glioblastoma tumorigenesis [121].

#### 4.2.3. USP21

The ubiquitin specific peptidase 21 (USP21) is the only centrosome and microtubule-associated DUB and localizes at the basal bodies in ciliated cells [73]. USP21 activity leads to the stabilization of many substrates, such as the pluripotency factor Nanog and the Mitogen-activated protein kinase kinase 2 (MEK2), a member of MAPK signaling cascade, thus sustaining stemness and cell proliferation, respectively [122,123]. Heride et al. described that USP21 positively regulates HH signaling either acting on the formation of primary *cilium* or altering GLI1 transcription activity [124], without excluding the interplay between these two mechanisms (Figure 5A). The authors demonstrated that USP21 and GLI1 form a complex and, together with PKA, colocalize at the centrosome in U2OS cells. Indeed, USP21 recruits GLI1 close to active PKA thus stimulating GLI1 phosphorylation [124,125]. Both depletion and overexpression of USP21 can hinder HH signaling, highlighting its regulatory role in the modulation of this pathway [124].

#### 4.2.4. USP37

Ubiquitin specific peptidase 37 (USP37) mainly localizes in the cytoplasm [126] and it has been initially described as a potent regulator of cell cycle at the G1/S transition, due to its ability to stabilize cyclin A. [127,128]. Moreover, USP37 is involved in the regulation of the stemness marker Nanog, of the EMT transcription factor Snail and of the oncoprotein C-Myc [129,130]. Qin et al. described the interplay among USP37 expression, the HH pathway, and EMT in breast cancer stem cells (BCSCs) [131]. In particular, they observed that genetic depletion of *USP37* in these cells induces the reduction of HH key components at protein level (such as SMO and GLI1) as well as of stem cell markers (i.e. ALDH1 and OCT4) [131]. In contrast, the activation of HH signaling induced by the agonist purmorphamine (PM) results in enhanced *USP37* gene expression that in turn stabilizes GLI1 (Figure 5A), and impacts on EMT in BCSBs [131]. These findings confirm the role of HH signaling in the maintenance of stem cells and EMT [132,133] and the implication of DUBs deregulations on these oncogenic processes [134,135]. Indeed, USP37 downregulation attenuates cell invasion and EMT markers expression by suppressing the HH pathway [131]. Moreover, in vivo xenograft mouse model of breast cancer showed that tumors resulting from *USP37* silenced cells are more sensitive to cisplatin, and have impaired HH target and stemness genes expression, together with lower proliferation ability compared to control group [131]. Overall these data indicate the relevance of USP37 in the regulation of breast cancer progression via the activation of the HH pathway.

#### 4.2.5. OTUB2

Ubiquitin thioesterase otubain-2 (OTUB2) is a deubiquitylating cysteine protease belonging to the ovarian tumor (OTU) superfamily of DUBs. Virus can encode DUBs to alter Ub-mediated host cell processes [136,137], and OTUB2 has been reported for its inhibitory activity on virus-triggered signaling through the deubiquitylation of TRAF3 and TRAF6 [138]. Further, OTUB2 affects DNA damage-dependent ubiquitylation, by protecting the polycomb molecule L3MBTL1 from RNF8-dependent degradation in an early phase of the DNA double-strand response (DDR) [139]. Recently, Li and co-workers described a new role for OTUB2 in the regulation of GLI2 stability (Figure 5A) [140]. In particular, the authors demonstrated the interaction between the two proteins and elucidated their interplay. The over-expression of OTUB2, but not of its catalytically inactive mutant C51A, protects GLI2 from proteasome-dependent degradation thus stabilizing and extending its half-life in U2OS cells [140]. Since HH signaling plays a relevant role in osteogenic differentiation during embryogenesis [141], Li et al. investigated the effects of *OTUB2* genetic depletion in mesenchymal stem cells (MSCs). They observed that HH stimulation promotes the expression of key drivers of osteoblast differentiation and bone formation, an effect that is inhibited in *OTUB2* knockdown condition. These findings outline OTUB2 as an agonist of HH signaling demonstrating its ability to stabilize GLI2 protein levels [140].

## 5. HH-Related DUBs: Inhibitors and Therapeutic Applications

Since the relevant role of DUBs in tumorigenesis, in the last decade many efforts have been devoted to the identification of selective DUBs inhibitors, demonstrating their therapeutic potential as anti-cancer agents [142,143,144]. DUBs that regulate key components of the HH pathway, such as USP7, USP8 and UCHL5/UCH37, are promising targets for the treatment of HH-dependent tumors. Their specific inhibitors with related chemical structures are summarized in Table 1 and Table 2, respectively.

USP7 is one of the most studied and best characterized DUB for its implication in different human diseases and in a wide spectrum of human cancers [145]. The first USP7 inhibitor was **HBX 41,108**, a cyanindenopyzazine derivative. HBX 41,108 acts on USP7 [146] through an uncompetitive reversible mechanism, binding this DUB after the formation of the enzyme/substrate complex. Although this molecule has shown selectivity towards USP7 in HCT116 human colon cancer cells, its weak activity against other related proteases limits the use for further pre-clinical studies [146]. 

Few years later, the same research team identified structurally distinct USP7 antagonists; among them, **HBX 19,818** exhibits an excellent selectivity for this DUB [147]. Reverdy and colleagues demonstrated that HBX 19,818 covalently binds the Cys223 located in the USP7 active site impairing cell proliferation and promoting apoptosis and cell cycle arrest in HCT116 cells. Of note, the viability of three cancer cell lines with different *p53* status is equally impaired by HBX 19,818 treatment, strongly suggesting that p53, one of the major USP7 target, is not required for the cellular response to USP7 inhibition. These findings suggest the existence of other USP7 substrates important for the proliferation of colon cancer cells [147]. To date, HBX 19,818 antitumor activity in vivo has not been yet described.

Subsequently, **P5091** a trisubstituted thiophene, and its analog **P22077** ((1-(5-((2,4-difluorophenyl)thio)-4-nitrothiophen-2-yl)ethanone)) have been identified as new USP7 inhibitors [148,149,150]. Although P22077 showed a cross-reactive effect against USP47, it has been used as USP7 antagonist in pre-clinical studies for neuroblastoma treatment. This compound significantly impairs tumor growth both in vitro and in vivo by inducing p53-mediated apoptosis or suppressing MYCN activity in *MYCN*-amplified neuroblastoma [151,152]. Interestingly, P22077 efficacy has been reported to suppress NSCLC and hepatocellular carcinoma tumor growth [153,154]. 

P5091 is one of the most well studied first-generation USP7 inhibitors, whose structure has been used as scaffold for chemistry optimization to develop new antagonists [150,155]. P5091 shows potent selective activity against USP7, inhibiting its ability to cleave high molecular weight poly-Ub chains in a dose-dependent manner. Chauhan and colleagues provided pre-clinical data on the anti-cancer efficacy of P5091 in multiple myeloma xenograft models. Interestingly, P5091 treatment impairs tumor growth by inducing apoptosis also in cells resistant to conventional and bortezomib therapies. All these evidence strongly support the clinical investigation of USP7 inhibitors, alone or in combination, as a valid therapeutic strategy for the treatment of multiple myeloma [150]. In addition, the potential therapeutic application of P5091 has also been reported for the treatment of various malignancies (Table 1) [156,157,158,159,160,161,162]. Notably, Zhan and colleagues showed that both P22077 and P5091 block proliferation and migration of MB cells, by reducing GLI proteins levels and inhibiting HH signaling [113].

Following advances in understanding the crystal structures of USP7, USP7-ligand complexes and its functional domains, several non-covalently binding USP7 inhibitors have been identified, including the 4-hydroxypiperidines **XL188** [163,164], **FT671** and **compound 4** [165,166], the 2-aminopyridine **GNE6640**, **GNE776** and the thiazole derivatives **C7** and **C19** [167,168]. Although these molecules show good potency and selectivity against USP7, further in vivo studies will be required to evaluate their therapeutic relevance in cancer treatment. 

Noteworthy, USP7 inhibitors have also been identified from natural sources, such as **spongiacidin C**, a pyrrole alkaloid obtained from the marine sponge *Stylissa massa* [169]. Despite biochemical assays show a good selectivity for USP7 for these compounds, their efficacy in cells remains to be determined.

Finally, in the last year, two new USP7 antagonists have been identified. **XL177A**, an analogue of **XL188**, is a small molecule that irreversibly inhibits USP7 with sub-nanomolar potency and selectivity, and whose effectiveness seems to be associated with *p53* mutational status in multiple cancer lineages [170]. On the contrary, **compound 41** is a reversible, highly potent, selective, and orally bioavailable USP7 inhibitor. In in vivo xenograft models of multiple myeloma, this molecule impairs the tumor growth of both p53 wild-type and mutant tumor cell lines, confirming that USP7 inhibition can suppress tumor growth affecting different pathways [171].

Currently only two molecules have been described as specific USP8 antagonists due to its pleiotropic function [85]. Colombo and colleagues identified the compound **9-ethyloxyimino-9H-indeno [1,2-b]pyrazine-2,3-dicarbonitrile** as the first specific USP8 inhibitor [172]. Subsequently, the effectiveness of this molecule has been reported to markedly decrease the in vitro and in vivo tumor growth of both gefitinib-sensitive and -resistant NSCLC cells [173]. 

The second USP8 inhibitor, **Ubv.8.2**, is an engineered ubiquitin variant identified to be a highly specific and potent inhibitor of this enzyme, showing the ability to occlude its Ub-binding site [174]. The only evidence of a potential anti-cancer activity for this molecule has been reported by MacLeod and co-workers, who demonstrated that the lentiviral expression of Ubv.8.2 leads to cell viability reduction in glioblastoma stem cell lines [175].

The potential application of DUBs inhibitors for the treatment of HH-related tumors includes also the exploitation of small molecules specific for those DUBs associated with proteasome, such as UCHL5 and USP14 [176]. These enzymes broadly act on substrates addressed to degradation machinery and represent the most investigated druggable DUBs. Indeed, their inhibition might have considerable effects on tumor cells, resulting in a less toxic strategy than targeting directly the proteasome complex [142].

Most of the UCHL5 inhibitors are also able to block the activity of USP14, known to modulate the HH pathway by controlling ciliogenesis [177]. Among the UCHL5/USP14 inhibitors identified so far, **b-AP15** (3,5-bis[(4-nitrophenyl)methylidene]-1-prop-2-enoylpiperidin-4-one) has been widely used for pre-clinical studies, exhibiting anti-cancer activity in both in vitro and in vivo models of different tumor types [178,179,180,181,182,183,184,185,186,187,188,189]. **VLX1570**, a derivative of b-AP15 [190], has been tested at pre-clinical level for the treatment of endometrial cancer, Ewing’s sarcoma and acute lymphoblastic leukemia [189,191,192]. Of note, this compound is the first DUB inhibitor to enter in clinical trial for the treatment of relapsed multiple myeloma (NCT02372240), although it has been suspended in Phase 1 for dose-limiting toxicity [190,193,194]. 

Despite UCHL5 and USP14 have very similar functions, selective inhibitors that individually target these enzymes have been identified. Among them, **IU1** (1-[1-(4-fluorophenyl)-2,5-dimethyl-1*H*-pyrrol-3-yl]-2-pyrrolidin-1-yl-ethanone) and its analogues are reversible small molecules that block specifically USP14 catalytic site and their antitumor effects have been recently tested in in vitro and in vivo studies for breast and lung cancers [195,196,197,198].

In the last decade, DUBs inhibitors are also emerged as versatile tools to define the structure and the cellular functions of these proteases. In addition, their potential application as anti-cancer agents stimulates the discovery of new and more specific antagonists. Research in this field could be particularly important, especially for those malignancies that have a complex biology such as HH-driven tumors.

## 6. Conclusions

The HH pathway is involved in the tumorigenesis of several malignancies and has emerged as a valid therapeutic target for anti-cancer therapy. At present, the main strategies to impair HH signaling are focused on inhibitors acting either on SMO or on GLIs, or through multi-targeting approaches working on both upstream and downstream levels [52,199,200]. A number of SMO antagonists have entered in clinical phases but only two of them, vismodegib and sonidegib, have been approved by the FDA for the treatment of BCC. Nevertheless, the response to SMO antagonists has been variable in other HH-dependent tumors such as MB, showing relapse due to lack of efficacy on SMO drug-resistant mutations and SMO-independent HH activation [201,202,203]. These limitations arose the need to develop alternative approaches. Even if GLI inhibitors have shown promising results in preclinical studies, few of them have entered in clinical studies, and only the Arsenic Trioxide (ATO) has been approved by FDA for the treatment of AML [33,34,51,200]. Currently, ATO is in several clinical trials for both solid tumors and hematological malignancies, but there are only preclinical studies for some HH-driven cancers such as MB. These results highlight that further efforts need to be spent on the development of more effective anti-cancer strategies for the treatment of HH-dependent tumors. In the last years, the possibility of hitting a tumorigenic pathway at multiple regulatory levels has emerging as a valid therapeutic frontier in the field of oncological research. The genetic and molecular heterogeneity of HH-driven malignancies stimulates the identification of novel molecular players of this pathway as potential druggable targets. In particular, ubiquitylation deeply rules HH signaling, and its pharmacological inhibition is an attractive tool to hinder this pathway at a further crucial level of regulation. In this regard, DUBs are emerging as interesting therapeutic targets in various HH-related tumors given their positive role in the control of the main performers of HH signaling. In addition to promoting the activity of SMO and GLIs, as here reviewed, DUBs affects HH signaling regulating ciliogenesis and the ciliary recruitment of HH regulatory proteins, as described for USP14 [177]. Moreover, multiple components of the HH pathway can be stabilized by the activity of DUBs. USP8, here presented for the function exerted on Smo, also regulates Itch, a HECT E3 ligase involved in GLI1 ubiquitylation. [204,205]. Notably, USP17, FAM/USP9X, and YOD1 have also been identified as modulators of Itch activity, enhancing its stability [206,207,208]. Recently, the involvement of βTrCP-bound deubiquitylase enzyme USP47 has been described in HH signaling. The interaction of the positive HH regulator ERAP1 with USP47 induces the degradation of βTrCP, thus protecting GLIs from βTrCP-dependent degradation and stimulating HH activity [209]. 

Increasing findings in this field of study highlight the interest in the development of more efficient and selective DUBs inhibitors for anti-cancer therapy, without affecting the fine physiologic balance of the Ub proteasome system that governs the proper functioning of all pathways, including HH.

## Figures and Tables

**Figure 1 cancers-12-01518-f001:**
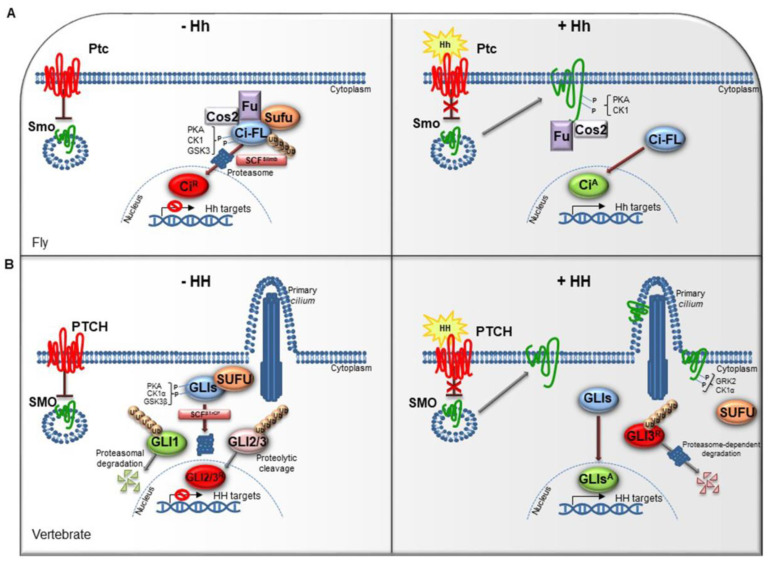
The Hedgehog signaling pathway. (**A**) The Hedgehog signaling pathway in fly. In absence of Hh, Ptc inhibits the localization of Smo on cell membrane. In the cytoplasm, Cos2, Fu and Sufu assemble in complex with Ci-FL protein, favoring its phosphorylation by PKA, CK1, and GSK3. This event induces the Ci-FL ubiquitylation by SCF^Slimb^ E3 ligase thus leading both to proteasome degradation and cleavage into truncated repressor form (Ci^R^). Ci^R^ blocks the transcription of Hh target genes. On the contrary, in the presence of Hh ligand, Ptc releases the inhibitory effect exerted on Smo which is activated by PKA and CK1 phosphorylation on the C-terminal domain, and then bound by Cos2 and Fu. These processes culminate in the Ci activation, promoting Hh transcription. (**B**) The Hedgehog signaling pathway in vertebrates. When the pathway is turned off, PTCH prevents the accumulation of SMO in the primary *cilium*. SUFU restrains GLI transcription factors in the cytoplasm where PKA, CK1α, and GSK3β kinases promote their phosphorylation. This process attracts the SCF^βTrCP^ E3 ligase that determines the processing of GLI2 and GLI3 (GLI2/3^R^) in their repressor forms and the proteasome-mediated degradation of GLI1. In presence of HH ligand, PTCH inhibition is relieved. SMO is accumulated in the primary *cilium* and activated by GRK2 and CK1α phosphorylation. GLI activator forms (GLIs^A^) translocate into the nucleus and induce the transcription of HH target genes.

**Figure 2 cancers-12-01518-f002:**
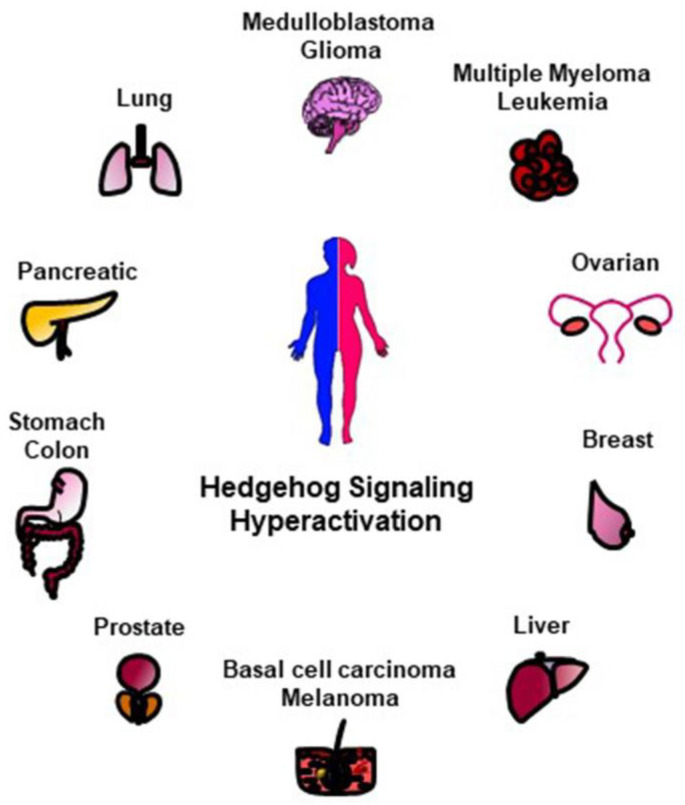
Schematic representation of HH-related tumors. The hyperactivation of HH signaling is involved in the tumorigenesis of several human malignancies here reported.

**Figure 3 cancers-12-01518-f003:**
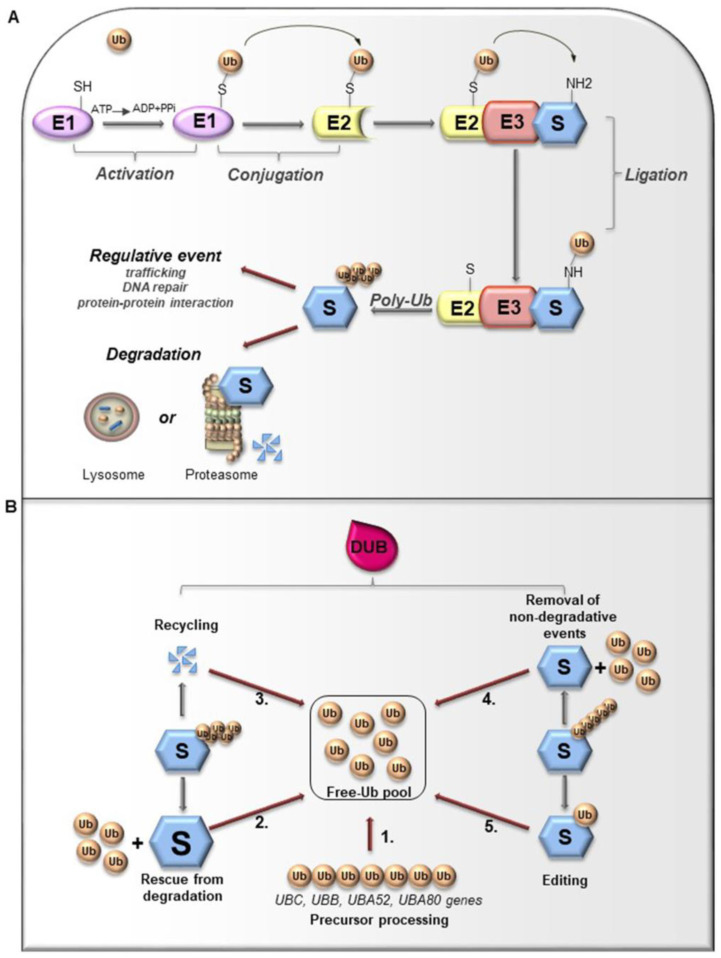
(**A**). Ubiquitylation processes. Ubiquitylation is a multi-step process that involves three enzymes: E1 (Ub-activating enzyme), E2 (Ub-conjugating enzyme) and E3 (Ub-ligase). Initially, Ub is linked to E1 through a high energy thioester bond. After, Ub activated by E1 is conjugated to a sulfhydryl group on E2 enzyme. Finally, E3 ligase specifically catalyzes the transfer of Ub from E2 to a Lys residue on a substrate protein. The formation of a poly-ubiquitin (poly-Ub) chain can lead the substrate toward a degradative or regulative pathway. (**B**). Deubiquitylation and DUBs function. Ubiquitylation can be reversed by deubiquitylating enzymes (DUBs) that hydrolyze the isopeptide or peptide bond, leading to Ub deconjugation from the ubiquitylated protein. DUBs have many functions. 1. Precursor processing: Ub is encoded by four genes and translated as a linear fusion protein consisting of multiple Ub copies, which require the cleavage by DUBs in order to generate free single Ub; 2. Rescue from degradation: DUBs can rescue protein from proteasomal or lysosome degradation; 3. Recycling: DUBs maintain Ub homeostasis preventing its degradation following substrate proteolysis; 4. Removal of non-degradative events: DUBs can remove Ub chains from substrates that are not committed to degradation; 5. Editing: DUBs can also affect the fate of ubiquitylated substrates by cleaving inter-Ub chains (switching from degradative to non-degradative ubiquitylation).

**Figure 4 cancers-12-01518-f004:**
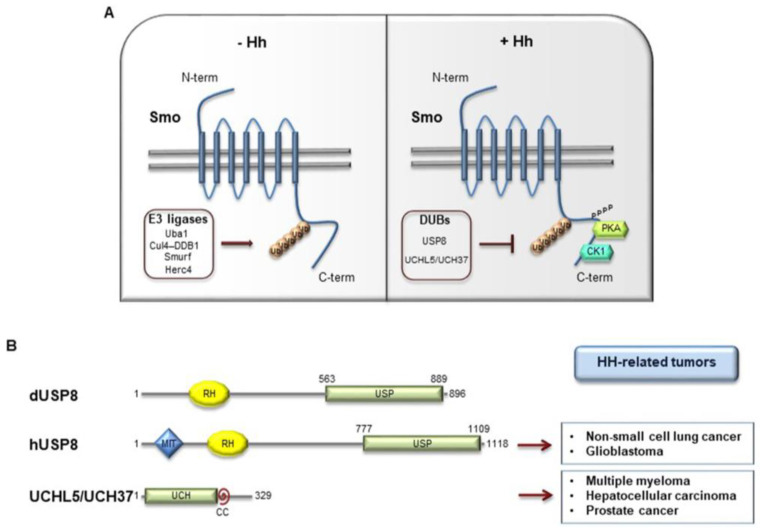
(**A**) E3-ubiquitin ligases and DUBs involved in Smo regulation. In *Drosophila*, when the Hh pathway is OFF, Smo is ubiquitylated by multiple E3 ligases (Uba1, Cul4-DDb1, Smurf, Herc4) that induce Smo lysosome or proteasome degradation. Conversely, when Hh signaling is activated, Smo is deubiquitylated by USP8 or UCHL5/UCH37 and then accumulated on cell surface, where it is active. This event is induced by PKA- and CK1-phosphorylation at the C-terminal region of Smo. (**B**) Structure of *Drosophila* dUSP8, human hUSP8, and UCHL5/UCH37. The boxes on the right indicate the main HH-related tumors in which USP8 and UCHL5/UCH37 are involved. USP: ubiquitin specific protease domain; MIT: microtubule interacting and trafficking molecule domain; RH: Rhodanese-like domain; CC: coiled coil domain.

**Figure 5 cancers-12-01518-f005:**
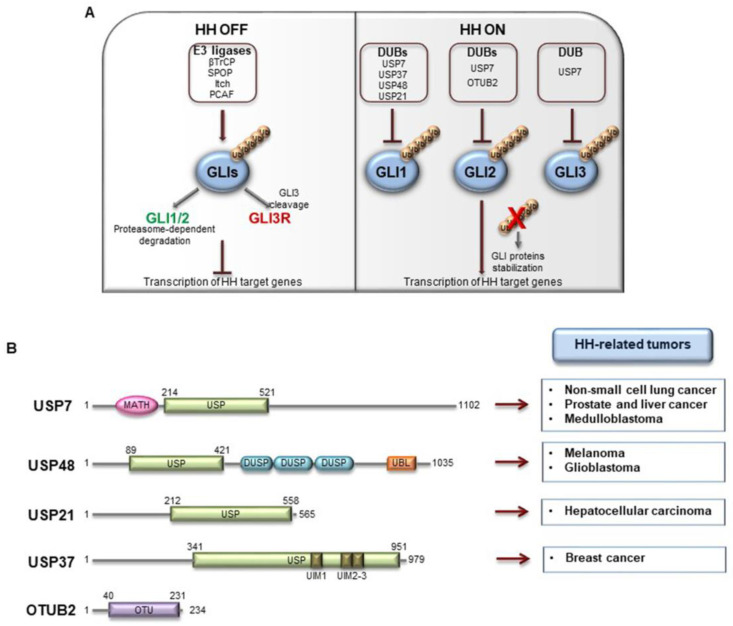
(**A**) E3-ubiquitin ligases and DUBs involved in GLI proteins regulation. In mammals, all GLI factors undergo ubiquitylation processing by different E3-ubiquitin ligase (βTrCP, SPOP, Itch, PCAF). When the pathway is OFF, SCF^βTrCP^ E3 ligase promotes the proteasome-dependent degradation of GLI1 and GLI2, and the proteolytic cleavage of GLI3 and to lesser extent GLI2 into the repressor forms. This modification blocks the transcriptional activity of GLIs. Ubiquitylation events are counteracted by the activity of DUBs (USP7, USP37, USP48, USP21, OTUB2) which remove Ub moieties from GLI factors. GLI proteins are then stabilized and can exert their activity promoting the expression of HH target genes. (**B**) Structure of human DUBs acting on GLI proteins. The boxes on the right indicate the main HH-related tumors in which these DUBs are involved. USP: ubiquitin specific protease domain; DUSP: dual-specificity phosphatase domain; UBL: ubiquitin-like domain; UIM: ubiquitin-interacting motif; OTU: ovarian tumour domain; MATH: meprin and TRAF homology domain.

**Table 1 cancers-12-01518-t001:** Inhibitors of HH-related DUBs.

DUB	Compound	Cancer Type (In Vitro and/or In Vivo Studies)	In Vivo Drug Administration	References
**USP7**	**HBX 41,108**	Colon cancer	-	[146]
	**HBX 19,818**	Colon cancer	-	[147]
	**P22077**	Neuroblastoma, Non-small cell lung cancer, Medulloblastoma, Hepatocellular carcinoma	Neuroblastoma orthotopic xenograft and hepatocellular carcinoma allograft mouse models: IP injection with P22077 dissolved in DMSO.	[113,148,151,152,153,154]
	**P5091**	Multiple Myeloma, Colorectal cancer, Prostate cancer, Ovarian cancer, Urothelial bladder cancer, Esophageal squamous cell carcinoma, Chronic lymphocytic leukemia, Medulloblastoma	Multiple myeloma xenograft mouse model: IV injection of P5091 dissolved in 4% NMP (N-methyl-2 Pyrrolidone), 4% Tween-80, and 92% H_2_O. Colorectal cancer xenograft mouse model: IP injection of P5091 dissolved in 4% NMP, 3%, Tween-80 and 20%, PEG400 in H_2_O. Esophageal squamous cell carcinoma xenograft mouse model: SC injection of P5091 dissolved in DMSO and 10% 2-hydroxypropyl-β-cyclodextrin (HPBCD).	[113,150,156,157,158,159,160,161,162]
	**XL188**	Ewing’s Sarcoma	-	[163,164]
	**FT671**	Colon cancer, Osteosarcoma Neuroblastoma, Multiple Myeloma	Multiple myeloma xenograft mouse model: oral gavage of FT671 dissolved in 10% DMA, 90% PEG400.	[165]
	**Compound 4**	Colon cancer, Breast cancer, Osteosarcoma, Prostate cancer	-	[168]
	**GNE6640, GNE6776**	Colon cancer, Osteosarcoma, Acute myeloid leukemia, Breast cancer	Acute Myeloid Leukemia and breast cancer xenograft mouse models: oral gavage of GNE-6776 dissolved in 0.5% methylcellulose, 0.2% Tween-80	[166]
	**C7, C19**	Colon cancer, Multiple Myeloma, Lung cancer	-	[167]
	**Spongiacidin C**	Not reported	-	[169]
	**XL177A**	Breast cancer, Ewing’s Sarcoma	-	[170]
	**Compound 41**	Multiple Myeloma, Small cell lung cancer	Multiple Myeloma and small cell lung cancer xenograft mouse models: oral gavage.	[171]
**USP8**	**9-ethyloxyimino-9H-indeno [1,2-b]pyrazine-2,3-dicarbonitrile**	Non-small cell lung cancer	Non-small cell lung cancer xenograft mouse model: IP injection.	[172,173]
	**Ubv.8.2**	Glioblastoma	-	[174,175]
**UCHL5** **USP14**	**b-AP15**	Squamous cell carcinoma, lung carcinoma, Colon cancer, Breast cancer, Acute myeloid leukemia, Multiple myeloma, Prostate cancer, Melanoma, Large B cell lymphoma, Neuroblastoma, Ewing’s sarcoma, Hepatocellular carcinoma	Squamous cell carcinoma, Lewis lung carcinoma, colon cancer, breast cancer xenograft and acute myeloid leukemia allograft mouse models: b-AP15 dissolved in Cremophor EL and polyethylene glycol 400 (1:1). Multiple myeloma xenograft mouse model: IV injection of b-AP15 dissolved in Cremophor EL/polyethylene glycol 400 (1:1). Prostate cancer xenograft mouse model: b-AP15 dissolved in a vehicle composed by DMSO, Cremophor and 0.85% NaCl at (1:3:6) ratio. Melanoma xenograft mouse model: IP injection of b-AP15 dissolved in 90:1:9 mix of Labrafil:Tween 80:DMA. Large B cell lymphoma xenograft mouse model: IP injection of b-AP15 dissolved in Cremophor.EL: PEG400: saline (2:2:4). Neuroblastoma and Ewing’s sarcoma xenograft mouse models: b-AP15 dissolved in DMSO.	[178,179,180,181,182,183,184,185,186,187,188,189]
	**VLX1570**	Ewing’s sarcoma, Multiple myeloma, Endometrial cancer, Acute lymphoblastic leukemia	Ewing’s sarcoma xenograft mouse model: IP injection of VLX1570 dissolved in DMSO. Multiple myeloma orthotopic xenograft mouse model: IV injection of VLX1570 dissolved in PEG:Chemophore:Tween (50:10:40). Relapsed Multiple myeloma patients (clinical trial NCT02372240): IV infusion.	[189,190,191,192,193]
**USP14**	**IU1** and **analogues**	Breast cancer, Lung cancer	-	[195,196,197,198]

IP: intraperitoneal; IV: intravenous; SC: subcutaneous.

**Table 2 cancers-12-01518-t002:** Chemical structures of DUBs inhibitors reported in Table 1.

Target	Compound	Structure	References
**USP7**	**HBX 41,108**	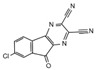	[146]
**HBX 19,818**	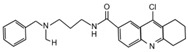	[147]
**P22077**	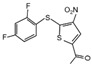	[148,149]
**P5091**	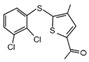	[150]
**XL188**	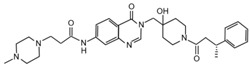	[163]
**FT671**	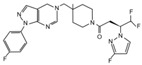	[165]
**Compound 4**	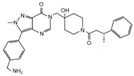	[168]
**GNE6640**	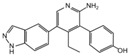	[166]
**GNE6776**	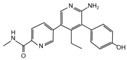	[166]
**C7**	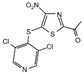	[167]
**C19**	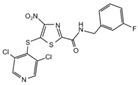	[167]
**Spongiacidin C**	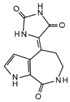	[169]
**XL177A**	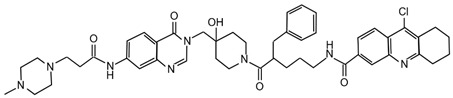	[170]
**Compound 41**	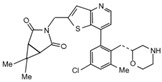	[171]
**USP8**	**9-ethyloxyimino-9H-indeno[1,2-b]pyrazine-2,3-dicarbonitrile**	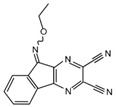	[172]
**UCHL5** **USP14**	**b-AP15**	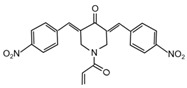	[178]
**VLX1570**	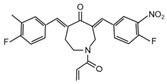	[189]
**USP14**	**IU1**	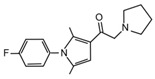	[195]

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
