# Peer review of "DUBs Activating the Hedgehog Signaling Pathway: A Promising Therapeutic Target in Cancer"

_cancers, 2020, doi:10.3390/cancers12061518_

Round 1
Reviewer 1 Report
- There are FDA approved drugs such as Vismodegib and Sonidegib which are SMO antagonists which focus on modulating SMO to directly to affect the Hedgehog (HH) pathway for cancer therapy. Similarly, Arsenic trioxide is an GL1 inhibitor approved for acute promyelocytic leukemia. There are antibodies targeting PTCH and GLI3 have also shown to affect the HH pathway. Itraconazole has also being tested for cancer therapy. The authors need to acknowledge existing drugs targeting HH pathway.
- The authors should explain the benefits behind targeting deubquitylation enzymes to affect HH pathway which are downstream targets when compared with existing direct approaches taken by SMO (Smoothened transmembrane protein), GL1 and SHH (Sonic hedgehog) inhibitors.
- Line 133, ‘In human, two E1s, around 30 E2s and over 600 E3s have been identified’. There is no citation for this statement.
- Line 204-206, ‘Studies aimed at defining the abundance of individual DUBs suggest that those with constitutive functions show high protein levels, while DUBs with peculiar roles display low expression’. There is no citation for this statement.
- Table 1 comprises the DUB inhibitors, however their delivery methods to target tumors also need to be described.
- Line 469, chemical name of 8.2Δ2 hasn’t been reported. An initial explanation of the chemical name or chemical class of these compounds is required.
- Need chemical structures or few examples of potent DUB inhibitors to understand structure activity relationships.
- Although figures explain the role of UB and DUB proteins and enzymes in hedgehog pathway, a final graphical representation explaining their role for cancer therapy or oncogenesis would give an overall summary of the article.
These typo’s need to be fixed
- Line 103, ‘non-small cells lung cancer’ should be corrected to ‘non-small cell lung cancer’.
- Line 120, ‘thus participating to the formation of’ should be corrected to ‘thus participating in the formation of’.
- Line 123, ‘Ubiquitylation dictates the fate and function of the most cellular proteins’ should be corrected to ‘Ubiquitylation dictates the fate and function of most cellular proteins’.
- Line 133, ‘In human, two E1s, around 30 E2s and over 600 E3s have been identified’ should be corrected to ‘In humans, two E1s, around 30 E2s and over 600 E3s have been identified’.
- Line 148-149, ‘often result in changes in the cellular localization and proteins activity’ should be corrected to ‘often result in changes in the cellular localization and protein activity’.
- Line 164, ‘binding domains and determining their accessibility to deubiquitylating enzymes’, should be corrected to ‘binding domains and determine their accessibility to deubiquitylating enzymes’.
- Line 166, Figure 3. (A). The substrate on the E2-E3-S complex is shown with a NH3 functional group. Should be corrected to NH2 functional group as the Ub attaches to the substrate Lys residue on the amino group (NH2 group). Figure 3. (B). Recycling is misspelled as Recicling.
- Line 176, ‘translated as a linear fusion consisting of multiple Ub copies’ should be corrected to ‘translated as a linear fusion protein consisting of multiple Ub copies’.
- Line 203, ‘For instance, OUT family exhibits’ should be corrected to ‘For instance, OTU family exhibits’.
- Line 207-208, ‘subsets of these protease’ should be corrected to ‘subsets of these proteases’.
- Line 243-244, ‘prevents Smo ubiquitylation and that is required for’ should be corrected to ‘prevents Smo ubiquitylation and is required for’.
- Line 289-292, ‘In mammals, HERC4 binds SMO and induces its degradation in human NSCLC. HERC4 knockdown activates HH signaling and promotes NSCLC cell proliferation thus standing as a tumor suppressor’ should be corrected to ‘In mammals, HERC4 binds SMO and induces its degradation. In human NSCLC, HERC4 knockdown activates HH signaling and promotes NSCLC cell proliferation thus standing as a tumor suppressor’.
- Line 299, ‘-independent’ should be corrected to ‘independent’.
- Line 355, ‘extents’ should be corrected to ‘extends’.
- Line 448, ‘and the its functional domains’ should be corrected to ‘and its functional domains’.
- Line 449, ‘including the 4-hydroxypeperidines’ should be corrected to ‘including the 4-hydroxypiperidines’.
- Line 465-466, ‘Colombo and colleagues identified the compound htyloxyimino-9H indeno [1,2-b]pyrazine-2,3-dicarbonitrile, should be corrected to ‘Colombo and colleagues identified the compound 9-ethyloxyimino-9H indeno [1,2-b]pyrazine-2,3-dicarbonitrile’.
- Line 473, ‘glioblastoma stem cells lines’ should be corrected to ‘glioblastoma stem cell lines’.
- Line 522, ‘write the manuscript’ should be corrected to ‘wrote the manuscript’.
Author Response
Response to Reviewer
REVIEW #1
- There are FDA approved drugs such as Vismodegib and Sonidegib which are SMO antagonists which focus on modulating SMO to directly to affect the Hedgehog (HH) pathway for cancer therapy. Similarly, Arsenic trioxide is an GL1 inhibitor approved for acute promyelocytic leukemia. There are antibodies targeting PTCH and GLI3 have also shown to affect the HH pathway. Itraconazole has also being tested for cancer therapy. The authors need to acknowledge existing drugs targeting HH pathway.
We have mentioned and discussed FDA approved HH antagonists and drugs targeting the GLI proteins, as requested (pages 4, 5 and 20).
- The authors should explain the benefits behind targeting deubquitylation enzymes to affect HH pathway which are downstream targets when compared with existing direct approaches taken by SMO (Smoothened transmembrane protein), GL1 and SHH (Sonic hedgehog) inhibitors.
We thank the reviewer for his/her suggestion. This aspect is discussed in the “Conclusion” section of the revised manuscript (page 20) .
- Line 133, ‘In human, two E1s, around 30 E2s and over 600 E3s have been identified’. There is no citation for this statement.
We have added the reference as suggested by Reviewer (refs 55, 56).
- Line 204-206, ‘Studies aimed at defining the abundance of individual DUBs suggest that those with constitutive functions show high protein levels, while DUBs with peculiar roles display low expression’. There is no citation for this statement.
We have corrected the sentence and added the reference as suggested by Reviewer (ref 70).
- Table 1 comprises the DUB inhibitors, however their delivery methods to target tumors also need to be described.
Delivery methods of the selected DUB inhibitors have been added in Table 1 as required by the Reviewer (revised Table 1).
- Line 469, chemical name of 8.2Δ2 hasn’t been reported. An initial explanation of the chemical name or chemical class of these compounds is required.
We thank the reviewer to have raised this point. Ubv.8.2 is an engineered ubiquitin variant identified to
be a highly specific and potent inhibitor of USP8 able to occlude the Ub-binding site of this
deubiquitylating enzyme. This explanation has been added in the revised manuscript.
- Need chemical structures or few examples of potent DUB inhibitors to understand structure activity relationships.
Chemical structures of selected DUBs inhibitors have been reported in new Table 2.
- Although figures explain the role of UB and DUB proteins and enzymes in hedgehog pathway, a final graphical representation explaining their role for cancer therapy or oncogenesis would give an overall summary of the article.
We thank the reviewer for this suggestion. Boxes that indicate the main HH-related tumors in which
the selected DUBs are involved have been added in Figure 4 and 5 (revised Figure 4 and Figure 5).
These typo’s need to be fixed
- Line 103, ‘non-small cells lung cancer’ should be corrected to ‘non-small cell lung cancer’.
- Line 120, ‘thus participating to the formation of’ should be corrected to ‘thus participating in the formation of’.
- Line 123, ‘Ubiquitylation dictates the fate and function of the most cellular proteins’ should be corrected to ‘Ubiquitylation dictates the fate and function of most cellular proteins’.
- Line 133, ‘In human, two E1s, around 30 E2s and over 600 E3s have been identified’ should be corrected to ‘In humans, two E1s, around 30 E2s and over 600 E3s have been identified’.
- Line 148-149, ‘often result in changes in the cellular localization and proteins activity’ should be corrected to ‘often result in changes in the cellular localization and protein activity’.
- Line 164, ‘binding domains and determining their accessibility to deubiquitylating enzymes’, should be corrected to ‘binding domains and determine their accessibility to deubiquitylating enzymes’.
- Line 166, Figure 3. (A). The substrate on the E2-E3-S complex is shown with a NH3 functional group. Should be corrected to NH2 functional group as the Ub attaches to the substrate Lys residue on the amino group (NH2 group). Figure 3. (B). Recycling is misspelled as Recicling.
- Line 176, ‘translated as a linear fusion consisting of multiple Ub copies’ should be corrected to ‘translated as a linear fusion protein consisting of multiple Ub copies’.
- Line 203, ‘For instance, OUT family exhibits’ should be corrected to ‘For instance, OTU family exhibits’.
- Line 207-208, ‘subsets of these protease’ should be corrected to ‘subsets of these proteases’.
- Line 243-244, ‘prevents Smo ubiquitylation and that is required for’ should be corrected to ‘prevents Smo ubiquitylation and is required for’.
- Line 289-292, ‘In mammals, HERC4 binds SMO and induces its degradation in human NSCLC. HERC4 knockdown activates HH signaling and promotes NSCLC cell proliferation thus standing as a tumor suppressor’ should be corrected to ‘In mammals, HERC4 binds SMO and induces its degradation. In human NSCLC, HERC4 knockdown activates HH signaling and promotes NSCLC cell proliferation thus standing as a tumor suppressor’.
- Line 299, ‘-independent’ should be corrected to ‘independent’.
- Line 355, ‘extents’ should be corrected to ‘extends’.
- Line 448, ‘and the its functional domains’ should be corrected to ‘and its functional domains’.
- Line 449, ‘including the 4-hydroxypeperidines’ should be corrected to ‘including the 4-hydroxypiperidines’.
- Line 465-466, ‘Colombo and colleagues identified the compound htyloxyimino-9H indeno [1,2-b]pyrazine-2,3-dicarbonitrile, should be corrected to ‘Colombo and colleagues identified the compound 9-ethyloxyimino-9H indeno [1,2-b]pyrazine-2,3-dicarbonitrile’.
- Line 473, ‘glioblastoma stem cells lines’ should be corrected to ‘glioblastoma stem cell lines’.
- Line 522, ‘write the manuscript’ should be corrected to ‘wrote the manuscript’.
All these typo’s have been corrected and the revised manuscript has been re-edited in several parts.

Reviewer 2 Report
In this manuscript, the authors reviewed the hedgehog (HH) pathway and deubiquitylase (DUBs) as a potential therapeutic target in cancer. The reviewer believes this review will help researcher in other fields or physician scientists to understand the overview of these pathways and possibilities. Several suggestions and discussion points are made to further improve on this work.
Major points
- The authors started with the summary of the HH pathway. Please shortly describe the source of the Hh ligand. In my understanding, both autocrine and paracrine have been reported in cancer. If it becomes clear, please compare it with normal tissues.
- Given that DUBs inhibitors should affect other substrates than Sumo and GLIs, please mention other substrates of the featured DUBs. While p53 was raised as a target in the section of DUBs inhibitors, broader description is preferred.
- Please refer to the approved drugs of the HH pathway, such as vismodegib, erismodegib or arsenic and explain the difference or advantage of the strategy to target DUBs.
Minor points
- All figures are helpful to understand. Small letters might be not readable. Please coordinate with the editorial staffs about the size of font.
- In Figure 2, some words should be revised. “Iperactivation”, “Ovaric”, “Prostatic”.
- “USP” in Figure 4B should correspond to “UCH” in the manuscript (line 268). Please match words.
Author Response
Response to Reviewer
REVIEW #2
In this manuscript, the authors reviewed the hedgehog (HH) pathway and deubiquitylase (DUBs) as a potential therapeutic target in cancer. The reviewer believes this review will help researcher in other fields or physician scientists to understand the overview of these pathways and possibilities. Several suggestions and discussion points are made to further improve on this work.
We thank the Reviewer for these positive general comments on our manuscript.
Major points
1. The authors started with the summary of the HH pathway. Please shortly describe the source of the Hh ligand. In my understanding, both autocrine and paracrine have been reported in cancer. If it becomes clear, please compare it with normal tissues.
We thank the reviewer for his/her suggestion. This aspect has been added in the paragraph “1. The HH signaling pathway and tumorigenesis: an overview” (pages 3 and 4, refs 35-44).
2. Given that DUBs inhibitors should affect other substrates than Sumo and GLIs, please mention other substrates of the featured DUBs. While p53 was raised as a target in the section of DUBs inhibitors, broader description is preferred.
We thank the reviewer for his/her suggestion. The main substrates of the featured DUBs are reported in the paragraph 4. “Oncogenic DUBs involved in the regulation of the HH pathway” of the revised manuscript.
3. Please refer to the approved drugs of the HH pathway, such as vismodegib, erismodegib or arsenic and explain the difference or advantage of the strategy to target DUBs.
We thank the reviewer for his/her suggestion. This aspect is discussed in the “Conclusions” section of the revised manuscript (page 20).
Minor points
1. All figures are helpful to understand. Small letters might be not readable. Please coordinate with the editorial staffs about the size of font.
We thank the reviewer for this suggestion. The size of font has been increased.
2. In Figure 2, some words should be revised. “Iperactivation”, “Ovaric”, “Prostatic”.
We apologize for these spelling mistakes. Figure 2 has been revised as suggested, and the revised manuscript has been re-edited in several parts.
3. “USP” in Figure 4B should correspond to “UCH” in the manuscript (line 268). Please match words.
Figure 4 has been corrected as suggested.

Reviewer 3 Report
In the review article entitled “DUBs activating Hedgehog signaling pathway: a 2 promising therapeutic target in cancer” Bufalieri and collaborators discuss the post translational role of ubiquination in the contest of Hedgehog (Hh) signaling. In particular, they cover the concept of deubiquitylase (DUBs) proteins associated to Hh pathway, their implication in tumorigenesis, and also their potential as drug target. In addition, the authors present recent insights into the development of selective DUB inhibitors.
This review is very through and comprises well designed figures that complement and integrate nicely the text. The manuscript is suitable for publication upon addressing few minor concerns.
Minor revisions:
Line 55: please change restraints into restrains.
Line 63: “The 62 full-length Ci protein is proteolytically processed, by the Skp1-Cullin1-Slimb […]” Please change into “The 62 full-length Ci protein is proteolytically processed by the Skp1-Cullin1-Slimb […]”
Line 66-68: can the authors briefly add the homology between the different HH ligands and summarize their main sites of expression?
Line 522: please change write into wrote
Author Response
Response to Reviewer
REVIEW #3
In the review article entitled “DUBs activating Hedgehog signaling pathway: a 2 promising therapeutic target in cancer” Bufalieri and collaborators discuss the post translational role of ubiquination in the contest of Hedgehog (Hh) signaling. In particular, they cover the concept of deubiquitylase (DUBs) proteins associated to Hh pathway, their implication in tumorigenesis, and also their potential as drug target. In addition, the authors present recent insights into the development of selective DUB inhibitors.
This review is very through and comprises well designed figures that complement and integrate nicely the text. The manuscript is suitable for publication upon addressing few minor concerns.
We thank the Reviewer for having appreciated the importance and soundness of our manuscript.
Minor revisions:
Line 55: please change restraints into restrains.
Line 63: “The 62 full-length Ci protein is proteolytically processed, by the Skp1-Cullin1-Slimb […]” Please change into “The 62 full-length Ci protein is proteolytically processed by the Skp1-Cullin1-Slimb […]”
Line 66-68: can the authors briefly add the homology between the different HH ligands and summarize their main sites of expression?
Line 522: please change write into wrote
All minor points have been addressed. The homology between the different HH ligands and their main sites of expression have been included in paragraph “1. The HH signaling pathway and tumorigenesis: an overview” of the revised manuscript (pages 2 and 3 and ref 7).

Round 2
Reviewer 1 Report
Authors addressed my comments with appropriate information.